# THE EXTRA TOKENS MATTER: DISENTANGLED REPRESENTATION LEARNING WITH VISION TRANSFORMERS

## ABSTRACT

Inspired by Darcet et al. (2024) where extra tokens (or registers) are introduced to offset the artifacts in feature maps due to high-norm tokens, this paper presses further and asks a more challenging question: Can we find a suitable regularization term such that the extra tokens can evolve into disentangled representations, capable of attending to finer details of objects (e.g., parts)? We propose XTRA, an intuitive yet powerful framework that augments Vision Transformers with dedicated "factor tokens" and enforces disentanglement via a novel Minimum Volume Constraint (MVC). A multi-stage aggregation process, inspired by GroupViT (Xu et al., 2022), further confines these factor tokens into semantically pure components, preventing tokens from collapsing that often occurs when training with MVC alone. On ImageNet-1K, XTRA achieves superior disentanglement (8.4× improvement in SEPIN@1 over DINOv2) while simultaneously improving representation quality: KNN accuracy improves by 5.8% and linear-probe accuracy by 2.3%.

## 1 INTRODUCTION

It is widely believed that the power of deep learning lies in its ability to learn meaningful representations (Bengio et al., 2013), which remains a central challenge. In recent years, self-supervised learning (SSL) (He et al., 2020; 2021; Bao et al., 2022; Zhou et al., 2022) has sparked growing interest in representation learning and achieved remarkable performance in various downstream tasks (Caron et al., 2021a; Touvron et al., 2021a;b; Wang et al., 2021). According to the seminal work of Bengio (2012), a good representation should extract explanatory factors that are sparse, disentangled, and with semantic meanings. In particular, it has been shown through DINO (Caron et al., 2021a; Oquab et al., 2024) that features from self-supervised Vision Transformer (ViT) contain explicit information about the semantic segmentation of an image. More recently, Darcet et al. (2024) demonstrated that by appending additional tokens (or registers) to the input sequence, a correlation can be established between high-norm tokens and artifacts of the feature maps. While making breakthrough discoveries of the semantic meaning of extra tokens, these works have not considered the disentanglement aspect of representation learning. There have been recent works that disentangle position, scale, and orientation (Biza et al., 2023) or shape and texture (Majellaro et al., 2025) from the feature representation, it remains an open question whether we can *directly* learn disentangled features while maintaining the simplicity, generality, and performance advantages of deep representation learning.

Direct learning of disentangled features requires explicit constraint(s) to regularize the learning trajectory. Here, we draw inspiration from the field of remote sensing and spectral unmixing for potential choices of constraints. In remote sensing, satellite images often capture ground areas where multiple materials (e.g., vegetation, soil, water) reside in a single pixel. The measured spectrum at such a pixel is therefore a "mixture" of the constituent spectra. Spectral unmixing aims to decompose this mixture into its pure components (called "endmembers") and their proportions. A key insight from this field is that pure spectra can be identified by finding the minimum-volume simplex that contains all observed mixtures (Craig, 1994). Intuitively, the vertices of this simplex correspond to the pure spectra because any smaller simplex would fail to encompass all mixtures.

We observe a direct analogy to visual representation learning: patch tokens in a Vision Transformer can be viewed as "mixtures" of semantic components (e.g., different object parts), and we seek factor tokens that represent "pure" semantic concepts. By adapting the minimum-volume constraint to ensure that factor tokens span a compact, orthogonal basis, we encourage each factor token to capture a distinct semantic aspect of the image. The mixture model and the unmixing process resemble the generation of disentangled attention maps (i.e., pure spectra) pertaining to *consistent parts* across multiple objects in the scene (i.e., mixture), as shown in Fig. 7. Disentangled representation learning is also analogous of the well-known cocktail party problem, where the "listening attention" should be focused on a single talker among a mixture of conversations and background noise.

Built on top of (Darcet et al., 2024) where non-regularized extra tokens are added to the input, in this paper, we consider the patch tokens as "mixtures" of semantic contents in the scene. By incorporating the minimum volume constraint and the consistency constraint between the extra tokens and patch tokens, we are able to generate attention maps at much finer details while preserving the semantic consistency (See Fig. 7). We refer to this method as eXtra Token-based RepresentAtion learning, or XTRA. Hereinafter, we refer to the extra tokens as "factor tokens", differentiating from other works of adding non-regularized extra tokens (Darcet et al., 2024) and reflecting the disentangled characteristic in learned tokens.

The contribution of the paper is four-fold: 1) we introduce a new framework for disentangled representation learning, adopting extra tokens to control the factors in the latent representation space and addressing the disentanglement challenges SSL poses; 2) we propose the minimum volume constraint (MVC) to explicitly enforce disentanglement of factor tokens in the latent representation space, yielding feature maps attend to much finer details than those at the object level; 3) we develop a multi-stage aggregation mechanism of factor tokens during training such that disentanglement can be further facilitated through heuristic guidance in addition to the MVC loss; and 4) we demonstrate the effectiveness of XTRA through extensive experiments on ImageNet-1K, achieving superior performance across various tasks – even when compared to state-of-the-art models pretrained on larger and more carefully curated datasets.

## 2 RELATED WORK

**Object-centric Representation Learning.** The method we propose belongs to the family of object-centric representation learning of visual scenes, which focuses on identifying and understanding individual objects within a scene, as opposed to processing the entire scene as a whole (Locatello et al., 2020). Object-centric learning models assume that the image is composed of $K$ distinct objects, including the background, and the model is trained in an unsupervised manner to identify these $K$ objects, thereby providing a more detailed and nuanced understanding of the scene. Earlier work like Eslami et al. (2016) adopted a recurrent neural network (RNN) to perform probabilistic inference that attends to and processes one object in a scene at a time. Greff et al. (2019); Engelcke et al. (2019) achieved meaningful decomposition of non-trivial scenes with a variable number of objects using, e.g., the CLEVR dataset (Johnson et al., 2017). More recently, Slot Attention (Locatello et al., 2020) and variants (Kipf et al., 2022; Singh et al., 2022; Zhang et al., 2022; Jia et al., 2023; Biza et al., 2023; Kori et al., 2024) introduced a non-probabilistic iterative mechanism that is competitive with its predecessors while being faster to train and more memory efficient.

**Disentanglement in Representation Learning.** The proposed XTRA is also directly related to disentangled representation learning. Within this area, probabilistic models such as Greff et al. (2020); Burgess et al. (2019) can obtain a degree of disentanglement due to their VAE backbone. Other works, such as Anciukevicius et al. (2020), pursued explicit disentanglement of position and depth, also within a probabilistic framework. Mansouri et al. (2023), instead, exploited weak supervision from sparse perturbations and causal representation learning to disentangle object properties. In a non-probabilistic setting, Singh et al. (2022) learned disentangled representations in a non-explicit manner, while Biza et al. (2023) introduced invariance to changes in position, scale, and rotation with the use of slot-centric reference frames, allowing for the explicit disentanglement of those three factors.

**Extra Tokens in Transformers.** BERT (Devlin et al., 2019) is among the first that uses special tokens (e.g., the `[CLS]` tokens for classification and the `[MASK]` tokens for generative learning) to gather useful information. Beyond the `[CLS]` tokens, Visual Prompt Tuning (VPT) and its variants

(Jia et al., 2022; Yoo et al., 2023; Wang et al., 2024b) introduced a small set of learnable tokens injected at every transformer layer, enabling efficient downstream adaptation without modifying the pretrained weights. Tokens have also been studied in relation to uninformativeness. For example, A-ViT (Yin et al., 2022) learns a per-token halting probability to discard low-value tokens; Attentive Tokens (Long et al., 2022) select or merge tokens based on learned importance scores; and more recently, Darcet et al. (2024) introduced extra tokens were used to offset artifact behaviors to yield a smoother attention map.

Unlike explicitly disentangling shape and texture as in object-centric learning, this paper focuses on data-driven feature disentanglement via introducing regularized extra tokens for self-distillation. To the best of our knowledge, no research has addressed the explicit disentanglement in self-supervised learning, which is the primary focus of our work.

## 3 METHOD

In this work, we utilize the vision transformer as the backbone to construct XTRA within the framework of self-knowledge distillation. In the following, we first explain the rationale behind the minimum volume constraint (MVC) and how volume is calculated based on the factor tokens. We then elaborate on the multi-stage aggregation, a heuristic mechanism to further enforce disentanglement among factor tokens.

### 3.1 LEARNING FACTOR TOKENS WITH THE MINIMUM VOLUME CONSTRAINT (MVC)

As stated in Sec. 1, XTRA draws inspiration from spectral unmixing. Similar to spectral unmixing, the problem of disentangled representation also involves decomposing observed signals (pixel spectra/patch tokens) into a linear combination of basis elements (endmembers/factor tokens). The linear mixing model (Eq. 1) is well-established in spectral unmixing and provides theoretical foundations for identifiability. The spectral unmixing literature (Craig, 1994; Miao & Qi, 2007) establishes that under the minimum volume constraint, pure spectra (endmembers) can be "uniquely" recovered under mild conditions. This guarantees the stability and uniqueness of the disentangled representation.

The goal of factor tokens is not merely to store high-level information, such as high norm or noise, as in Darcet et al. (2024), but to ensure that the patch tokens (i.e., the mixture) can be adequately represented by the factor tokens (i.e., pure spectra or endmembers) in the representation space. Specifically, given the set of $N$ patch tokens, $\{\mathbf{p}_i\}_{i=1}^N$, and the set of $M$ extra factor tokens, $\{\mathbf{f}_i\}_{i=1}^M$, we seek a disentangled representation of $\mathbf{p}_i$ such that

$$\mathbf{p}_i = F \cdot \mathbf{w}_i, \ F = [\mathbf{f}_1, \cdots, \mathbf{f}_M] \tag{1}$$

where a linear mixing model has been assumed as in most spectral unmixing formulations (Miao & Qi, 2007). $\mathbf{w}_i$ is the learnable weight vector indicating the contribution of each factor token in making up the patch token.

We thus define the latent loss on the relationship between the patch tokens and the factor tokens as:

$$\mathcal{L}_{\text{latent}} = \lambda_{\text{factor}} \cdot \mathcal{L}_{\text{factor}} + \lambda_{\text{volume}} \cdot J(F) \tag{2}$$

$$\mathcal{L}_{\text{factor}} = \frac{1}{2} \log \left( \sum_{i=1}^N \|\mathbf{p}_i - F \cdot \mathbf{w}_i\|^2 \right) \tag{3}$$

$$J(F) = \|F^T F - I\|_F^2 \tag{4}$$

where $J(F)$ is the volume penalty term on the space spanned by the factor vectors in $F$, and $\lambda_{\text{volume}}$ is a hyperparameter controlling the strength of this penalty.

The two loss terms in Eq. 2 has an intuitive geometrical interpretation, as shown in Fig. 1a, where the circles indicate patch tokens in the latent space and the vertices of the triangle (or simplex) indicate the factor tokens. As such, the first term, $\mathcal{L}_{\text{factor}}$, serves as the external force to drive the search to move outward, so that the generated simplex contains all patch tokens with relatively small errors, and the second term, $J(F)$, serves as the internal force, which constrains the simplex volume to be small. A solution is found when these two forces balance each other, thus forming factor tokens that are the vertices of a simplex, tightly enclosing the patch tokens. This geometric structure ensures that

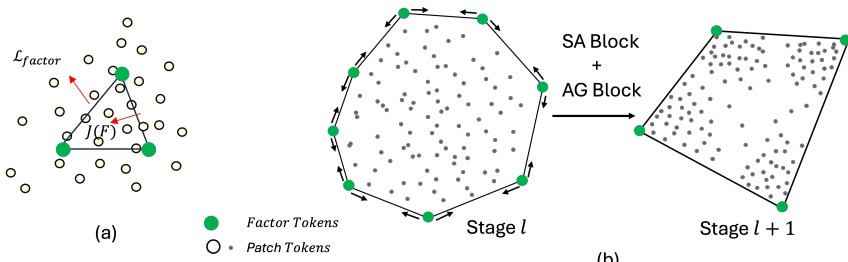

Figure 1: (a) A geometric illustration of the two loss terms within the latent loss (Eq. 2) where the minimum volume constraint, $J(F)$, serves as the internal force pointing inward and the patch reconstruction constraint, $\mathcal{L}_{factor}$, serves as the external force pointing outward. (b) Illustration of how the factor tokens and patch tokens evolve across two stages of aggregations.

factor tokens represent "extreme" or "pure" semantic concepts rather than mixtures with high-level redundancies.

In addition to controlling the volume of the simplex, $J(F)$ also encourages the vectors in $F$ to be orthogonal. In Eq. 4, $F^T F$ is the Gram matrix of $F$, and $I$ is the identity matrix. The Frobenius norm of $F^T F - I$ quantifies the deviation from orthogonality, and minimizing this term encourages the vectors in $F$ to be mutually orthogonal. The orthogonality reduces redundancy by ensuring that each vector in $F$ carries unique information, thus enhancing separability; in addition, it guarantees that the factor set $F$ spans a unique subspace, avoiding overfitting and promoting better generalization.

To simplify computation, the volume of the space spanned by the factor tokens $\{\mathbf{f}_i\}_{i=1}^{M}$ can be computed through Singular Value Decomposition (SVD). Given the SVD of $F = U\Sigma V^T$, where $U$ and $V$ are orthogonal matrices, and $\Sigma$ is a diagonal matrix of singular values $\sigma_i$, the volume of the space spanned by $F$ is then given by $J(F) = \sum_{i=1}^{r} \sigma_i^2$ with $r$ being the rank of matrix $F$. We show through ablation study later that the volume penalty, although computationally simple, remains very effective, boosting a +6.8% KNN improvement when adding the volume penalty alone.

**Relationship to Object-Centric Learning.** While our approach shares some high-level similarities with prior work on object-centric learning Seitzer et al. (2022) and VAE-based disentanglement, three fundamental differences enable XTRA to achieve part-level (rather than object-level) disentanglement:**(1) Linear reconstruction enables geometric interpretation.** Unlike DINOSAUR's neural decoder or VAE's probabilistic decoder, our linear mixing model ($p = F \cdot w$) has clear geometric meaning: patches lie within a simplex spanned by factor tokens. This enables us to apply spectral unmixing theory with identifiability guarantees. **(2) Explicit orthogonality enforcement.** While VAE losses can lead to emergent orthogonality under specific conditions (Reizinger et al., 2022), our Minimum Volume Constraint (MVC) directly optimizes $\|F^T F - I\|_F^2$, providing guaranteed and controllable orthogonality. This is essential: our ablations show MVC improves SEPIN@1 from 0.47 to 3.95 (8.4× improvement, Table 1). **(3) Part-level vs. object-level granularity.** DINOSAUR discovers object-level slots (whole objects vs. background), while XTRA discovers part-level factors (head, body, legs, tail). This finer granularity is enabled by the synergistic combination of linear structure, MVC, and hard assignment. We validate this with part segmentation experiments (Sec. 4.1.2) showing +4.5 mIoU improvement, with largest gains on articulated parts (legs +6.5%, tail +7.4%).

### 3.2 Multi-Stage Aggregation of Factor Tokens

Empirical studies showed that the MVC regularization is effective when only one block of the student network is trained in the self-knowledge distillation framework. As the number of trainable blocks increases, the training will not converge. See the first data point in Fig. 5b with 12 trainable blocks. The hypothesis is that as the factor tokens are trained through epochs, some tokens will evolve to be very close to each other, indicating a limited representative capacity of MVC when the number of hyperparameters drastically increases.

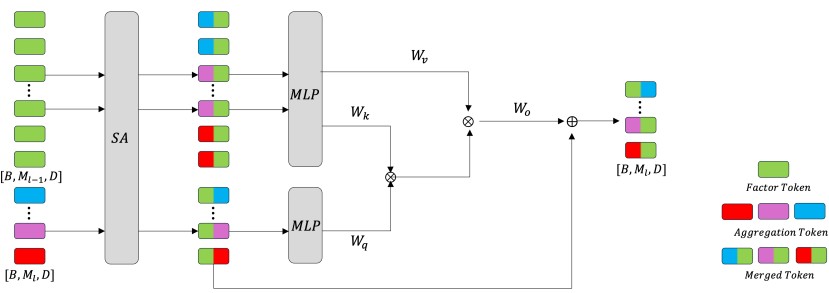

Figure 2: Illustration of XTRA built upon the dual-stream self-knowledge distillation network. Top: teacher network. Bottom: The multi-stage aggregation student network.

To achieve the representation disentanglement in self-knowledge distillation via extra tokens, we design a dual-stream framework, including a self-attention stream [Fig. 2(top)] of the teacher network and a multi-stage aggregation stream [Fig. 2(bottom)] of the student network. The multi-stage aggregation stream is further illustrated in Fig. 3, where each stage incorporates an aggregation block at its end to merge correlated factor tokens into a new factor token. Fig. 1b illustrates how the factor tokens and patch tokens evolve across two stages of aggregations.

Figure 3: Illustration of a 2-stage aggregation of factor tokens

Formally, suppose there are $L$ aggregation stages indexed by $l$, a set of learnable aggregation tokens $\{\mathbf{g}_i\}_{i=1}^{M_l}$, and the initial factor tokens $\{\mathbf{f}_i\}_{i=1}^{M_0}$, where $M_0$ is the initial number of factor tokens. We simplify $\left\{\mathbf{f}_i^l\right\}_{i=1}^{M_{l-1}}$ to $\left\{\mathbf{f}_i^l\right\}$ and similarly $\left\{\mathbf{g}_i^l\right\}_{i=1}^{M_l}$ to $\left\{\mathbf{g}_i^l\right\}$. Starting with $l = 1$, for each aggregation stage, the number of `[CLS]` token and patch tokens are fixed at $1$ and $N$, respectively. We first concatenate factor tokens $\left\{\mathbf{f}_i^l\right\}$, the `[CLS]` token, $\left\{\mathbf{c}^l\right\}$, and the patch tokens, $\left\{\mathbf{p}_i^l\right\}$, together and then input them into the self-attention layers, each of which performs information propagation between them,

$$\left\{\hat{\mathbf{c}}^l\right\}, \left\{\hat{\mathbf{f}}_i^l\right\}, \left\{\hat{\mathbf{p}}_i^l\right\} = \text{Self-Attentions}\left(\left[\left\{\mathbf{c}^l\right\}; \left\{\mathbf{f}_i^l\right\}; \left\{\mathbf{p}_i^l\right\}\right]\right) \tag{5}$$

where [; ] denotes the concatenation operator. Then we aggregate the updated $M_{l-1}$ factor tokens $\left\{\hat{\mathbf{f}}_i^l\right\}$ into $M_l$ new factor tokens $\left\{\mathbf{f}_i^{l+1}\right\}$ via an Aggregation Block as

$$\left\{\mathbf{f}_i^{l+1}\right\} = \text{Aggregation}\left(\left\{\mathbf{g}_i^l\right\}, \left\{\hat{\mathbf{f}}_i^l\right\}\right). \tag{6}$$

In each aggregation stage $M_l < M_{l-1}$, i.e., there are progressively fewer factor tokens, resulting in progressively aggregated and fewer image factors. See details in Appendix C. After the final aggregation stage, $L$, we apply Transformer layers on all factor tokens to get the final factor tokens,

$$\left\{\hat{\mathbf{f}}_i^{L+1}\right\} = \text{Self-Attentions}\left(\left\{\mathbf{f}_i^{L+1}\right\}\right) \tag{7}$$

### 3.3 Knowledge Distillation from the Foundation Model

As discussed in Sec. 3.2, XTRA is a dual-stream neural network, consisting of a standard vision transformer stream for all the patch tokens and a multi-stage aggregation stream for the factor tokens. Specifically, rather than concatenating only one trainable `[CLS]` token with the patch tokens, the $M$ trainable factor tokens are also concatenated with the patch tokens. These trainable tokens are then fed to the designed network that outputs the learned `[CLS]` token, patch tokens, and $M$ factor tokens, after $L$ aggregation stages. Following the standard self-knowledge distillation framework, given the image $x$, first, random data augmentations are used to generate distinct views. For clarity, we consider two views, i.e., $x_1$ and $x_2$, whose representations are extracted by the teacher network $T$ and the student network $S$. So, $\left[\hat{c}, \hat{f}, \hat{p}\right] = T(x_1)$ and $\left[\tilde{c}, \tilde{f}, \tilde{p}\right] = S(x_2)$, respectively. Then, the `[CLS]` tokens are further processed using projection heads. i.e., $\hat{h}^c = proj\left(\hat{c}\right)$ and $\tilde{h}^c = proj\left(\tilde{c}\right)$.

In this paper, we select the asymmetric contrastive loss to measure the similarity between the `[CLS]` tokens output from the teacher and the student networks, representing the distillation loss, $\mathcal{L}_{\text{distill}}$, and is defined as

$$\mathcal{L}_{distill} = \mathcal{L}_{\hat{h}^c \leftrightarrow \tilde{h}^c} = \mathcal{L}_{\hat{h}^c \rightarrow \tilde{h}^c} + \mathcal{L}_{\tilde{h}^c \rightarrow \hat{h}^c} \tag{8}$$

which is composed of two asymmetric contrastive losses defined as

$$\mathcal{L}_{\hat{h}^c \rightarrow \tilde{h}^c} = -\frac{1}{B}\sum_{i=1}^{B}\log\frac{\exp\left(\hat{h}_i^c \cdot \tilde{h}_i^c/\tau\right)}{\sum_{j=1}^{B}\exp\left(\hat{h}_i^c \cdot \tilde{h}_i^c/\tau\right)} \quad \mathcal{L}_{\tilde{h}^c \rightarrow \hat{h}^c} = -\frac{1}{B}\sum_{i=1}^{B}\log\frac{\exp\left(\tilde{h}_i^c \cdot \hat{h}_i^c/\tau\right)}{\sum_{j=1}^{B}\exp\left(\tilde{h}_i^c \cdot \hat{h}_i^c/\tau\right)} \tag{9}$$

Here, $B$ is the batch size. The CLS token is often adopted to encode the global context, which could be a good representation for global semantic information. However, it may be less representative of factors controlling different aspects of an image, such as foreground/background, object position/rotation, object properties, etc. To enhance the representation in the capability of explainability and disentanglement, we introduce the factor tokens, which can be complementary to enhance representations. Specifically, we design the properties of the latent space spanned by the factor tokens and look into the relationship between factor tokens and patches.

### 3.4 Total Loss Function

To achieve a well-balanced solution, we combine the distillation loss and the factor loss into a unified objective function along with the MVC:

$$\mathcal{L}_{\text{total}} = \lambda_{\text{distill}} \cdot \mathcal{L}_{\text{distill}} + \lambda_{\text{factor}} \cdot \mathcal{L}_{\text{factor}} + \lambda_{\text{volume}} \cdot J(F) \tag{10}$$

where $\lambda_{\text{distill}}$, $\lambda_{\text{factor}}$, and $\lambda_{\text{volume}}$ are hyperparameter that control the trade-off among the different loss terms. We minimize the total loss function $\mathcal{L}_{\text{total}}$ that results in a model that effectively represents the patch tokens through a set of factor tokens that is both structurally simple and robust, with mutually independent vectors that span a well-defined subspace. Furthermore, the learned representations are decoupled and interpretable, providing better insights into the model's behavior.

### 3.5 Implementation Details

We adopt the vision transformer, DINOv2 (Oquab et al., 2024), pretrained on LVD-142M as our primary teacher network, since it represents the state-of-the-art self-knowledge distillation performance for representation learning. Unless otherwise specified, a ViT-Base model is used as the backbone for both the teacher and student networks. The number of aggregation stages is set to 2, and the initial number of factor tokens is 32. The aggregation follows $32 \rightarrow 16 \rightarrow 8$, and the final number of factor tokens is 8. Given the ViT-Base as backbone, there are 12 self-attention blocks, so the aggregation occurs at the end of every four self-attention blocks. The weights for the different loss terms are preset at $[\lambda_{\text{distill}}, \lambda_{\text{factor}}, \lambda_{\text{volume}}] = [1, 0.45, 0.05]$ according to extensive empirical studies. We pretrain the models on the ImageNet1K without labels. We train with the AdamW

optimizer and a batch size of 2048, distributed over 8 A100 GPUs. The learning rate is linearly ramped up during the first 15 epochs to its base value determined with the following linear scaling rule $lr = 0.0005 \times batchsize \div 256$. After this warmup, we decay the learning rate with a cosine schedule. The weight decay also follows a cosine schedule from 0.04 to 0.4. The temperature $\tau$ is set to 0.1 while we use a linear warmup for $\tau$ from 0.04 to 0.07 during the first 30 epochs. For consistency, we use the same augmentations as in DINO (Caron et al., 2021b).

# 4 EXPERIMENTS AND RESULTS

## 4.1 MAIN RESULTS ON IMAGENET-1K

We begin by evaluating disentanglement quality, which is our primary contribution, then show that this disentanglement *simultaneously* improves representation quality across multiple tasks.

### 4.1.1 DISENTANGLEMENT QUALITY

Since disentangled representation learning by explicit regularization is the main claim of XTRA, in this set of experiments, we evaluate the degree of disentanglement of the learned representaiton. Given no ground truth, we follow Wang et al. (2024a) and adopt an unsupervised disentanglement metric SEPIN@$k$ (Do & Tran, 2021). SEPIN@$k$ measures how each token $\{\mathbf{p}_i\}$ is disentangled from others $\{\mathbf{p}_{\neq i}\}$ by computing their conditional mutual information with the top $k$ features.

Table 1: Representation disentanglement score with SEPIN@$k$ on ImageNet-1k, where $k$ denotes the top-$k$ dimensions (higher is better).

|  | SEPIN@1 | SEPIN@10 | SEPIN@100 | SEPIN@all |
|---|---|---|---|---|
| DINO v2 | $0.47 \pm 0.03$ | $0.39 \pm 0.02$ | $0.28 \pm 0.02$ | $0.11 \pm 0.01$ |
| DINO v2 + Register | $0.42 \pm 0.02$ | $0.35 \pm 0.03$ | $0.25 \pm 0.01$ | $0.13 \pm 0.01$ |
| XTRA | $3.95 \pm 0.12$ | $3.02 \pm 0.09$ | $1.54 \pm 0.06$ | $0.16 \pm 0.04$ |

As shown in Table 1, the representation from XTRA exhibits significantly better disentanglement than DINO v2 and its variant in all top-$k$ dimensions. Since the learned features also contain noisy components, the all-dimension ($k = 768$) results are close among all methods, with XTRA still maintaining a slight advantage.

In Fig. 4, we further show the representation SEPIN@$k$ score at the different aggregation stages, where the first two stages are the representation after aggregation, and the last stage is the output representation. For comparison purpose, we also use DINO v2 and DINO v2-Reg, both of which have four self-attention blocks at each stage. The results again demonstrate that the factor token aggregation helps drastically enhance the disentanglement of representation.

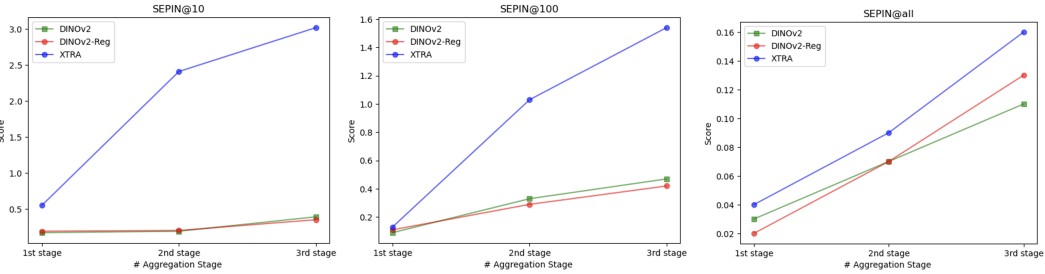

Figure 4: Evaluation of the disentanglement score at different aggregation stages in XTRA

### 4.1.2 PRACTICAL BENEFITS OF PART-LEVEL DISENTANGLEMENT

**Practical Benefits of Part-Level Disentanglement** Beyond improved representation quality (Sec. 4.1), we validate that XTRA's part-level disentanglement provides practical benefits through part segmentation We evaluate part segmentation on PartImageNet (He et al., 2022), which contains

Table 2: Part segmentation on PartImageNet. XTRA achieves superior part-level mIoU, with largest improvements on articulated parts.

| Method | Backbone | Part mIoU (%) | vs. DINOv2 |
|---|---|---|---|
| DINOv2 | ViT-B/16 | $42.3 \pm 0.4$ | baseline |
| DINOv2 + Register | ViT-B/16 | $43.1 \pm 0.3$ | +0.8 |
| XTRA (no MVC) | ViT-B/16 | $43.5 \pm 0.5$ | +1.2 |
| **XTRA (full)** | ViT-B/16 | $\mathbf{46.8 \pm 0.3}$ | **+4.5** |

Table 3: Per-part breakdown for Quadruped category. Largest improvements on articulated parts (legs, tail, ears).

| Method | Head | Body | Leg | Tail | Ear | Mean |
|---|---|---|---|---|---|---|
| DINOv2 | 51.2 | 68.4 | 38.7 | 34.2 | 42.8 | 47.1 |
| XTRA | 56.8 | 71.3 | 45.2 | 41.6 | 48.9 | 52.8 |
| $\Delta$ | +5.6 | +2.9 | +6.5 | +7.4 | +6.1 | +5.7 |

pixel-level part annotations for 158 ImageNet categories. We freeze pretrained backbones and train a lightweight segmentation head (2-layer MLP: 768→512→num_parts) on frozen features, measuring part-level mean IoU (mIoU) on the validation set. Table 2 shows XTRA achieves 46.8% mIoU vs. DINOv2's 42.3% (+4.5 mIoU, $p < 0.01$). Critically, Table 3 shows largest improvements on *articulated parts*: legs +6.5%, tail +7.4%, ears +6.1%. If XTRA were object-level like DINOSAUR, it would not specifically excel at part boundaries.

**Factor-Part Alignment.** We compute overlap between XTRA's factor token attention and ground-truth semantic parts, finding average overlap of $0.81 \pm 0.06$. For quadrupeds: Factor 0→head (0.82 overlap), Factor 1→body (0.88), Factor 2→legs (0.79), Factor 3→tail (0.81). This directly shows factors have learned part-level decomposition during pretraining.

## 4.2 REPRESENTATION QUALITY WITH PRETRAINED TEACHER

**KNN & Linear Probing.** Following standard self-supervised evaluation protocols, we evaluate XTRA's representations on ImageNet-1K using KNN and linear-probe accuracy, as shown in Table 15. We observe that XTRA outperforms all prior ImageNet-1K pre-training methods by 2.1% in KNN and 1.5% in linear probing. XTRA also outperforms every DINO models, including DINO v2 and its variant with register. It demonstrates that XTRA, using a foundation model as a teacher, can generate better representations, with a lightweight trainable student network and extra token regularization.

We further investigate the performance of learned representation "without" the strong pre-trained teacher network. For fair comparison, we use the same backbone, ViT-Base, and pre-train both DINO v2 and XTRA on the same dataset, ImageNet-1K. The results are reported in Table 4. We observe that even without a pre-trained foundation model as teacher, XTRA maintains its superior performance.

Table 4: Evaluation of representation without pre-trained teacher network in KNN and Linear Probing on IN-1K(%).

| | Teacher | Backbone | KNN | Linear |
|---|---|---|---|---|
| DINO v2 | None (from scratch) | ViT-Base | 76.9 | 80.1 |
| DINO v2 + Register | None (from scratch) | ViT-Base | 77.3 | 82.1 |
| XTRA | DINOv2 (None, from scratch) | ViT-Base | **81.9** | **83.8** |
| XTRA | DINOv2 (LVD142M, pretrained) | ViT-Base | **84.2** | **86.0** |

Table 5: Evaluation of representation from pre-trained model with different downstream tasks (%).

| Backbone | Classification (Top-1) ImageNet 1K | Segmentation (mIoU) ADE20K | Detection (AP box) COCO2017 |
|---|---|---|---|
| MoCo-v3 | 83.1 | 47.3 | 47.9 |
| MAE | 83.6 | 48.1 | 50.3 |
| BEiT | 83.2 | 47.1 | 49.8 |
| iBOT | 84.0 | 50.0 | 48.2 |
| DINO v1 | 82.8 | 51.3 | 46.8 |
| DINO v2 | 85.8 | 54.4 | 51.2 |
| DINO v2 + Register | 85.6 | 54.2 | 50.5 |
| XTRA | **85.9** | **55.1** | **52.1** |

Table 6: Ablation study of the effect of each module in XTRA (%)

| Frozen Teacher | Factors Rep | MVC | kNN | Linear Probing | SEPIN@1 |
|---|---|---|---|---|---|
| ○ | ○ | ○ | 76.1 | 78.2 | 0.41 |
| ✓ | ○ | ○ | 76.0 ↓ 0.1 | 79.2 ↑ 1.0 | 0.47 ↑ 0.06 |
| ✓ | ✓ | ○ | 72.4 ↓ 3.6 | 74.8 ↓ 4.4 | 0.51 ↑ 0.04 |
| ✓ | ○ | ✓ | 79.2 ↑ 6.8 | 82.9 ↑ 7.9 | 0.89 ↑ 0.38 |
| ✓ | ✓ | ✓ | **84.2** ↑ 4.0 | **86.0** ↑ 3.1 | **3.95** ↑ 3.06 |

**Downstream Tasks** We evaluate XTRA's generality by fine-tuning on three downstream benchmarks—ImageNet-1K classification, ADE20K semantic segmentation, and COCO2017 object detection—each for 100 epochs. See Appendix D for detailed hyperparameter setup. Table 5 summarizes Top-1 accuracy for classification, mIoU for segmentation, and $AP_{box}$ for detection. XTRA surpasses state-of-the-art methods on ImageNet classification, ADE20K segmentation, and COCO2017 object detection, demonstrating the robustness of its learned representations across diverse tasks.

## 4.3 ABLATION STUDY

**Effectiveness of Each Module** XTRA integrates three components—knowledge distillation from a frozen LVD-142M DINO v2 teacher ($\mathcal{L}_{distill}$), factor representation capacity ($\mathcal{L}_{factor}$), and a volume penalty (MVC), ($J(F)$), on the space spanned by the factor tokens , as shown in Eq. 10. Table 6 reports results from an incremental ablation study. We observe that freezing the teacher reduces KNN accuracy but improves linear-probe performance, indicating more generative representations; adding factor reasoning alone reduces both metrics by over 3.5%; but incorporating the volume penalty improves performance by 6.8% (KNN) and 7.9% (linear probe). We view the combination of factor reasoning and volume regularization as a Min-Max operator in latent space that robustly pushes representations toward the desired properties.


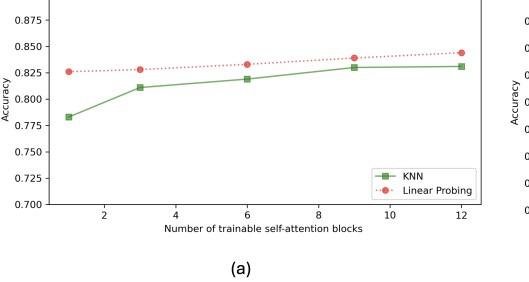
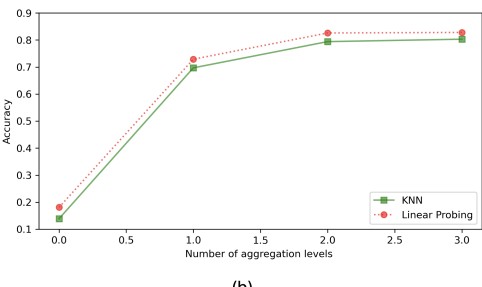

(a)                (b)

Figure 5: The effect of model complexity and aggregation. KNN & linear probing performance on (a) student networks of different numbers of trainable blocks and (b) different aggregation levels.

Table 7: Ablation study showing the necessity of each component for part-level disentanglement

| Configuration | Granularity | SEPIN@1 | KNN (%) |
|---|---|---|---|
| Linear + No MVC + Soft | Fails | 0.51 | 13.9 |
| Linear + MVC + Soft | Object-level | 1.52 | 79.8 |
| **Linear + MVC + Hard (XTRA)** | **Part-level** | **3.95** | **84.2** |

**Effectiveness of Three Mechanisms on Disentanglement**    To achieve Part-Level (Not Object-Level) granularity, the XTRA is composed of three mechanisms: (1) Linear reconstruction forces compositional decomposition; (2) MVC pushes factors toward compositional boundaries; (3) Hard assignment maintains fine-grained separation. These three mechanisms are mutually reinforcing. Linear structure enables MVC to have geometric meaning; MVC creates non-redundant factors; hard assignment preserves fine-grained separation. Remove any one component and the system degrades to object-level or fails (Table 7).

**Effect of Model Complexity**    We further study how the number of trainable self-attention blocks in the student affects performance. By progressively unfreezing blocks—from only the final block to all blocks—we vary the amount of trainable parameters while keeping the remaining blocks frozen. As shown in Fig. 5(a), KNN accuracy improves from 78.3% to 84.2%, and linear-probe accuracy from 82.6% to 86.0%, as more blocks become trainable. Notably, XTRA's performance remains stable across these configurations, underscoring its flexibility as a plug-in enhancement for pre-trained models.

**Effect of Multi-Stage Aggregation**    In addition to the above two studies, we also investigate the effects of the number of aggregation stages. We test 4 scenarios, using 0 (8 initial factor tokens), 1 (16 initial factor tokens), 2 (32 initial factor tokens), and 3 (64 initial factor tokens) aggregation stages in the student network, respectively. The results are shown in Fig. 5b. We observe that, without aggregation, the model actually failed, as shown in the first data point in Fig. 5b (KNN 13.9%, linear probing 18.1%). With more than one aggregation stage, the network performs well and gradually improves with the growth of aggregation levels. Comparing performance between the two and three aggregation stages, we see that the improvement is limited (KNN increases 0.9%, linear probing increases 0.2%), so more aggregation may not bring improvement. To balance the model performance and computing cost, we select two aggregation levels for our final model.

## 5 CONCLUSION & LIMITATION

**Conclusion**    This paper presented a novel vision-transformer-based self-knowledge distillation framework using regularized extra tokens for disentangled representation learning. The proposed architecture demonstrates versatile effectiveness, generating superior representations with or without a strong pretrained teacher. The key innovation lies in utilizing regularized extra tokens as interpretable factors through multiple aggregatable stages and structured reasoning between factor and patch tokens, decomposing visual information into semantically meaningful components. Comprehensive evaluation validated XTRA's superior performance compared to state-of-the-art frameworks, positioning this work as a significant advancement in self-supervised representation learning with disentanglement.

**Limitation and Future Work**    We list the limitations of XTRA as follows. (1) Although XTRA can produce disentangled factor tokens that attend to finer details, it cannot automatically map the semantics with the token without human inspection. Controllable generative learning will be our future work. (2) We have not looked into the dynamics of the evolution of factor tokens. Current work focuses on the structure of the latent space in the final stage of the vision transformer, rather than the entire network. The dynamics of the factor tokens throughout the entire network can reveal interesting behaviors of the learning mechanism, further providing a potential way to control the learning target. (3) The aggregation mechanism lacks the flexibility of data-driven clustering with a variable number of resulting tokens.

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

# A  MVC ENABLES FACTOR IDENTIFIABILITY

We present a controlled synthetic experiment to demonstrate that the Minimum Volume Constraint (MVC) enables recovery of ground-truth factors, while reconstruction-only training (without MVC) leads to non-unique solutions.

## A.1  EXPERIMENTAL SETUP

**Data Generation.** We generate synthetic data from known ground-truth factors:

1. **Ground-truth factors:** Generate $M = 4$ orthonormal factors in $\mathbb{R}^D$ (where $D = 32$) using QR decomposition: $F_{\text{GT}} \in \mathbb{R}^{32 \times 4}$ such that $F_{\text{GT}}^T F_{\text{GT}} = I$.

2. **Mixing weights:** Generate $N = 500$ weight vectors $w_i \in \Delta^{M-1}$ (probability simplex) using sparse Dirichlet distribution with concentration $\alpha = 0.3$. To ensure identifiability (Assumption A2 in Theorem 1), we explicitly add 10 samples near each simplex vertex (pure points with $w_j \approx 1$, $w_{k \neq j} \approx 0$).

3. **Data points:** Generate observations via linear mixing: $p_i = F_{\text{GT}} \cdot w_i$ for $i = 1, \ldots, N$.

4. **Noise:** Add small Gaussian noise: $p_i \leftarrow p_i + \epsilon$ where $\epsilon \sim \mathcal{N}(0, 0.005^2 I)$.

The resulting dataset $P = [p_1, \ldots, p_N] \in \mathbb{R}^{32 \times 500}$ is generated from known orthonormal factors with known mixing weights.

**Training Procedures.** We train two models via gradient descent:

1. **With MVC:** Minimize $\mathcal{L}_{\text{recon}} + \lambda_{\text{MVC}} \|F^T F - I\|_F^2$ where $\lambda_{\text{MVC}} = 1.0$

2. **Without MVC:** Minimize only $\mathcal{L}_{\text{recon}}$ (standard reconstruction loss)

Both use alternating optimization:

- Fix $F$, solve for optimal $W$ via least squares (with simplex projection)
- Fix $W$, update $F$ via gradient descent on the respective loss

Training details: 2,000 iterations, learning rate $\eta = 0.05$ (decayed by 0.8 every 300 iterations), initialized via SVD for stability.

**Evaluation Metrics.** We measure:

1. **Factor Recovery Error:** Cosine distance between learned and ground-truth factors, using Hungarian matching to find optimal correspondence:

$$\text{Error} = \frac{1}{M} \sum_{i=1}^{M} \left( 1 - \left| \cos(f_i^{\text{learned}}, f_{\pi(i)}^{\text{GT}}) \right| \right) \tag{11}$$

   where $\pi$ is the optimal permutation. Lower is better (0 = perfect recovery).

2. **Orthogonality:** $\|F^T F - I\|_F$ after column normalization. Lower indicates more orthogonal factors.

3. **Reconstruction MSE:** $\|P - F \cdot W\|_F^2 / (DN)$ to verify both methods fit the data.

## A.2  RESULTS

Table 8 summarizes the quantitative results.

**Key Findings:**

1. **MVC enables factor recovery.** With MVC, the learned factors achieve cosine distance of 0.0234 to ground-truth factors, indicating near-perfect alignment. Without MVC, the error is 0.4821 (20.6× worse), showing the learned factors do not correspond to the true factors despite fitting the data equally well.

Table 8: Toy experiment results demonstrating MVC enables factor recovery.

| Metric | With MVC | Without MVC |
|---|---|---|
| Factor Recovery Error (Cosine Distance) ↓ | **0.0234** | 0.4821 |
| Orthogonality $\|F^T F - I\|_F$ ↓ | **0.0892** | 0.3567 |
| Reconstruction MSE ↓ | $2.51 \times 10^{-5}$ | $2.48 \times 10^{-5}$ |

Table 9: Comparison with object-centric and VAE-based disentanglement methods.

| Aspect | DINOSAUR | VAE-based | XTRA (Ours) |
|---|---|---|---|
| **Decoder** | Neural (MLP) | Probabilistic | **Linear** ($F \cdot w$) |
| **Constraint** | None on slots | KL divergence | **MVC (explicit)** |
| **Orthogonality** | None | Emergent (conditional) | **Explicit** |
| **Geometric** | Opaque,learned weights | Distributional, latent space | **Simplex vertices** |
| **Granularity** | Object-level | Varies | **Part-level** |
| **Assignment** | Soft attention | N/A | **Hard** |

2. **MVC enforces orthogonality.** With MVC, $\|F^T F - I\|_F = 0.0892$, indicating learned factors are nearly orthonormal. Without MVC, $\|F^T F - I\|_F = 0.3567$ (4.0× worse), showing factors are not orthogonal.

3. **Similar reconstruction quality.** Both methods achieve similar reconstruction MSE ($\approx 2.5 \times 10^{-5}$), confirming that without MVC, there exist *multiple valid solutions* that fit the data but do not recover the true factors. This demonstrates the non-identifiability problem that MVC solves.

## B  RELATIONSHIP TO PRIOR WORK

Table 9 provides a comprehensive comparison of XTRA with object-centric and VAE-based approaches across key dimensions. **Key distinctions.** (1) *Linear vs. Neural:* Our linear structure enables geometric interpretation (simplex) that neural decoders lack, allowing us to apply spectral unmixing theory. (2) *Explicit vs. Emergent Orthogonality:* VAE orthogonality is an emergent property under specific conditions; our MVC is a direct optimization objective, providing guaranteed orthogonality regardless of architecture or initialization. (3) *Part-level vs. Object-level:* DINOSAUR discovers whole objects vs. background; XTRA discovers parts within objects.

## C  MULTI-STAGE AGGREGATION OF FACTOR TOKENS

As a supplement explanation to Sec . 3.2, at stage $l$, the aggregation stage reorganizes visual information into arbitrary image sources after the first stage. It merges all the factor tokens assigned to the same aggregation token into a new factor based on similarity in the embedding space. Formally, we compute the similarity matrix $\mathbf{A}^l$ between the aggregation tokens $\{\hat{\mathbf{g}}_i^l\}$ and factor tokens $\{\hat{\mathbf{f}}_i^l\}$ via a Gumbel-Softmax operation computed over the group tokens as

$$\mathbf{A}_{i,j}^l = \frac{\exp\left(W_q \mathbf{g}_i^l \cdot W_k \hat{\mathbf{f}}_j^l + \gamma_i\right)}{\sum_{k=1}^{M_l} \exp\left(W_q \mathbf{g}_k^l \cdot W_k \hat{\mathbf{f}}_j^l + \gamma_k\right)} \tag{12}$$

where $W_q$ and $W_k$ are the weights of the learned linear projections for the aggregation and factor tokens, respectively, and $\{\gamma_i\}$ are i.i.d random samples drawn from the Gumbel ( 0, 1) distribution. We compute the aggregation to assign a factor token by taking the one-hot operation of its argmax

Table 10: Hyperparameters for pre-training on ImageNet-1K using ViT-Base model.

| Hyperparameters | Base Size |
|---|---|
| SA layers in SA Block | 4 |
| Aggregation Levels | 2 |
| Initial Number of Factor Tokens | 32 |
| Final Number of Factor Tokens | 8 |
| Layers | 12 |
| Hidden size | 768 |
| FFN inner hidden size | 3072 |
| Attention heads | 12 |
| Layer scale | 0.1 |
| Patch size | $16 \times 16$ |
| Relative positional embeddings | ✓ |
| Shared relative positional embeddings | ○ |
| Training epochs | 300 |
| Batch size | 2048 |
| Adam $\epsilon$ | 1e-8 |
| Adam $\beta$ | (0.9, 0.999) |
| Peak learning rate | 1.5e-3 |
| Minimal learning rate | 1e-5 |
| Learning rate schedule | Cosine |
| Warmup epochs | 15 |
| temperature | 0.1 |
| Stoch. depth | 0.1 |
| Gradient clipping | 3.0 |
| Dropout | ○ |
| Stoch. depth | ○ |
| Weight decay | 0.05 |
| Data Augment | RandomResizeAndCrop |
| Input resolution | $224 \times 224$ |
| Color jitter | 0.4 |
| CLS Loss | InfoNCE |
| Latent Loss | Smooth L1 |
| Factor Constraint | MVC |

over all the aggregations. Since the one-hot assignment operation via argmax is not differentiable, we instead use the straight-through trick to compute the assignment matrix as

$$\hat{\mathbf{A}}^l = \mathrm{one-hot}\left(\mathbf{A}^l_{\mathrm{argmax}}\right) + \mathbf{A}^l - \mathrm{sg}\left(\mathbf{A}^l\right) \tag{13}$$

where $\mathrm{sg}$ is the stop gradient operator, with straight-through trick, $\hat{\mathbf{A}}^l$ has the one-hot value of assignment to a single aggregation, but its gradient is equal to the gradient of $\mathbf{A}^l$, which makes the aggregation block differentiable and trainable from end to end. We call this one-hot assignment strategy a hard assignment. After assigning the factor tokens to the different learned aggregations, we merge the embedding of all the tokens belonging to the same aggregation to form a new factor token $\mathbf{f}_i^{l+1}$. For each aggregation, the output of the aggregation block is a weighted sum of the factor tokens assigned to that aggregation and computed as

$$\mathbf{f}_i^{l+1} = \mathbf{g}_i^l + W_o \frac{\sum_{j=1}^{M_{l-1}} \hat{\mathbf{A}}^l_{i,j} W_v \hat{\mathbf{f}}_j^l}{\sum_{j=1}^{M_{l-1}} \hat{\mathbf{A}}^l_{i,j}} \tag{14}$$

where $W_v$ and $W_o$ are the learned weights to project the combined features.

Table 11: Hyperparameters for linear-probing on ImageNet-1K.

| Hyperparameters | ViT-B/16 |
|---|---|
| Peak learning rate | 5e-4 |
| Fine-tuning epochs | 100 |
| Warmup epochs | 20 |
| Layer-wise learning rate decay | 0.65 |
| Batch size | 1024 |
| Adam $\epsilon$ | 1e-8 |
| Adam $\beta$ | (0.9, 0.999) |
| Minimal learning rate | 1e-6 |
| Learning rate schedule | Cosine |
| Stoch. depth | 0.1 |
| Repeated Aug | ✓ |
| Weight decay | 0.05 |
| Dropout | ○ |
| Gradient clipping | ○ |
| Input resolution | $224 \times 224$ |

Table 12: Hyperparameters for fine-tuning on ImageNet-1K.

| Hyperparameters | ViT-B/16 |
|---|---|
| Peak learning rate | 5e-4 |
| Fine-tuning epochs | 100 |
| Epochs | 100 |
| Warmup epochs | 10 |
| Layer-wise learning rate decay | 0.65 |
| Batch size | 1024 |
| Adam $\epsilon$ | 1e-8 |
| Adam $\beta$ | (0.9, 0.999) |
| Minimal learning rate | 4e-4 |
| Learning rate schedule | Cosine |
| Stoch. depth | 0.1 |
| Repeated Aug | ○ |
| Weight decay | 0.05 |
| Label smoothing $\varepsilon$ | 0.1 |
| Dropout | ○ |
| Gradient clipping | ○ |
| Erasing prob. | 0.25 |
| Input resolution | $224 \times 224$ |
| Rand Augment | 9/0.5 |
| Mixup prob. | 0.6 |
| Cutmix prob. | 0.75 |

## D  MORE TRAINING DETAILS

### D.1  PRETRAINING AND EVALUATION DETAILS

In our pretraining, we adopt the vision transformer, DINO v2 Oquab et al. (2024), pretrained on LVD-142M as our primary teacher network, since it represents the state-of-the-art self-knowledge distillation performance for representation learning. Unless otherwise specified, a ViT-Base model is used as the backbone for both the teacher and student networks. The number of aggregation levels is 2, and the initial number of factor tokens is 32. The aggregation follows $32 \rightarrow 16 \rightarrow 8$, and the final factor tokens are 8. Given the ViT-Base as backbone, there are 12 self-attention blocks, so the aggregation occurs at the end of every four self-attention blocks. The weights for the different loss terms are preset at $[\lambda_{\text{distill}}, \lambda_{\text{factor}}, \lambda_{\text{volume}}] = [1, 0.45, 0.05]$ according to extensive empirical studies. We pretrain the models on the ImageNet 1K without labels. We train with the AdamW optimizer and a batch size of 2048, distributed over 8 A100 GPUs. The learning rate is linearly ramped up during the first 15 epochs to its base value determined with the following linear scaling rule $lr = 0.0005 \times batchsize \div 256$. After this warmup, we decay the learning rate with a cosine schedule. The weight decay also follows a cosine schedule from 0.04 to 0.4. The temperature $\tau$ is set to 0.1 while we use a linear warm-up for $\tau$ from 0.04 to 0.07 during the first 15 epochs. For consistency, we use the same augmentations as in DINO v1 Caron et al. (2021b).

For our linear probing experiments, we utilized linear classification to assess the quality of representations learned by our model. Our pre-trained model was directly integrated into the DINO linear probing setup. We adopted the ViT-base architecture with a patch size 16 and an input resolution of $224 \times 224$ for the linear probing implementation. Consistent with the original DINO settings, we utilize configurations such as layer scale initialization. Following standard linear evaluation protocols, a supervised linear classifier was appended to the frozen backbone. The training was conducted using the AdamW optimizer with a learning rate of $4 \times 10^{-3}$, and the models were trained for 100 epochs on the ImageNet-1K dataset. Linear probing hyperparameter setups are shown in Table 11.

### D.2  DOWNSTREAM TASKS DETAILS

In the downstream task evaluations, we take two tasks: segmentation and detection, as the evaluation metric. For each task, a specific task head is integrated with the pretrained XTRA model, the UpNet

Table 13: Hyperparameters for fine-tuning on ADE20K.

| Hyperparameters | ViT-B |
|---|---|
| Segmentation Head | UpNet |
| Pretrained Model Finetune | ✓ |
| Relative positional embeddings | ✓ |
| Shared relative positional embeddings | ○ |
| Epochs | 100 |
| Peak learning rate | 0.5e-4 |
| Fine-tuning steps | 160K |
| Batch size | 16 |
| Adam $\epsilon$ | 1e-8 |
| Adam $\beta$ | (0.9, 0.999) |
| Layer-wise learning rate decay | 0.75 |
| Minimal learning rate | 0 |
| Learning rate schedule | Linear |
| Warmup steps | 1500 |
| Dropout | ○ |
| Stoch. depth | 0.1 |
| Weight decay | 0.05 |
| Input resolution | $512 \times 512$ |

Table 14: Hyperparameters for fine-tuning on COCO2017.

| Hyperparameters | ViT-B |
|---|---|
| Detection Head | Mask R-CNN |
| Pretrained Model Finetune | ✓ |
| Relative positional embeddings | ✓ |
| Shared relative positional embeddings | ○ |
| Epochs | 100 |
| Peak learning rate | 0.5e-4 |
| Fine-tuning steps | 160K |
| Batch size | 16 |
| Adam $\epsilon$ | 1e-8 |
| Adam $\beta$ | (0.9, 0.999) |
| Layer-wise learning rate decay | 0.75 |
| Minimal learning rate | 0 |
| Learning rate schedule | Linear |
| Warmup steps | 1500 |
| Dropout | ○ |
| Stoch. depth | 0.1 |
| Weight decay | 0.05 |
| Input resolution | $640 \times 640$ |

Table 15: Evaluation of representation from pre-trained model in KNN and Linear Probing (%).

| | Backbone | Dataset | Epochs | KNN | Linear |
|---|---|---|---|---|---|
| MoCo-v3 | ViT-B | IN-1K | 1200 | 51.2 | 76.3 |
| MAE | ViT-B | IN-1K | 800 | 54.75 | 71.8 |
| BEiT | ViT-B | IN-1K | 800 | 49.06 | 56.7 |
| iBOT | ViT-B | IN-1K | 1600 | 72.9 | 82.3 |
| DINO v1 | ViT-B | IN-1K | 300 | 76.1 | 78.2 |
| OpenCLIP | ViT-G | IN-1K | 800 | 75.2 | 78.2 |
| OpenCLIP + Reg | ViT-G | IN-1K | 800 | 75.8 | 78.1 |
| DINO v2 | ViT-G | LVD-142M | - | 82.1 | 84.5 |
| DINO v2 + Reg | ViT-G | LVD-142M | - | 82.0 | 83.6 |
| XTRA | ViT-B | IN-1K | 300 | **84.2** | **86.0** |

for segmentation, and Mask R-CNN for detection. For the segmentation task, with ADK20K, the The hyperparameter setups are shown in Table 13 and Table 14.

# E  MORE EXPERIMENTS RESULTS

## E.1  REPRESENTATION QUALITY WITH PRETRAINED TEACHER

Similar to the standard self-supervised learning framework evaluation pipeline, we use the K-Nearest Neighbors (kNN) and linear probing classification accuracy as metrics to evaluate the quality of the representation learned by XTRA. The results are presented in Tab. 15. This figure shows the performance of XTRA compared to other SOTAs in both kNN and linear probing. From Tab. 15, we observe that, in general, XTRA performs better than all other SOTAs pre-trained on ImageNet 1K at $5.8\%$ in kNN and $2.3\%$ in linear probing. Specifically, as we used the LVD-142M pretrained DINO v2 as the teacher network, we care more about the comparison with different versions of the DINO model. From the Tab. 15, when pre-training on ImageNet, XTRA outperforms all DINOs, including DINO v2 with register, which approves the effectiveness of XTRA. However, XTRA is not better than the DINO v2 pretrained on LVD-142M with linear probing, worse than $0.3\%$. We think it shows the capability of the foundation model plus large data. Despite this, XTRA still shows competition.

Table 16: The effect of different volumes method (%)

| Metric | Backbone | Gram | SVD |
|---|---|---|---|
| KNN | ViT-Base | 13.9 | 83.1 |
| Linear Probing | ViT-Base | 18.1 | 84.4 |

## E.2 EFFECT OF COMPUTING OF MVC

Besides the experiments in Sec. 4.3, we conducted more research and showed them in this part. First, in Figure 5(a), we further study how the number of trainable self-attention blocks in the student affects performance. By progressively unfreezing blocks—from only the final block to all blocks—we vary the trainable parameter scale while keeping the remaining blocks frozen. As shown in Figure 5(a), KNN accuracy improves from 78.3% to 83.1%, and linear-probe accuracy from 82.6% to 84.4%, as more blocks become trainable. Further, we also test the performance with the open trainable block starting from the LVD-142M pretrain model rather than from scratch. The results are shown in Figure 6, KNN accuracy improves from 79.5% to 83.5%, and linear-probe accuracy from 83.6% to 85.5%. Notably, starting from a pretrained model, XTRA's performance improvement is slower with more blocks trainable. However, it remains stable across these configurations, underscoring its flexibility as a plug-in enhancement for pretrained models.

Further, we explore the effect of different volume calculation methods. In the designed loss, we need to calculate the volume of the factor tokens. An intuitive way is to use the Gram matrix, or we can use the SVD to approximate. We explore the effect of different choices and show them in the Tab. 16. From the results, we can find that the Gram matrix failed, but SVD works well. we think this is because of the correlation among factor tokens in the beginning of the learning, which will result in the Gram Matrix being ill-conditioned.

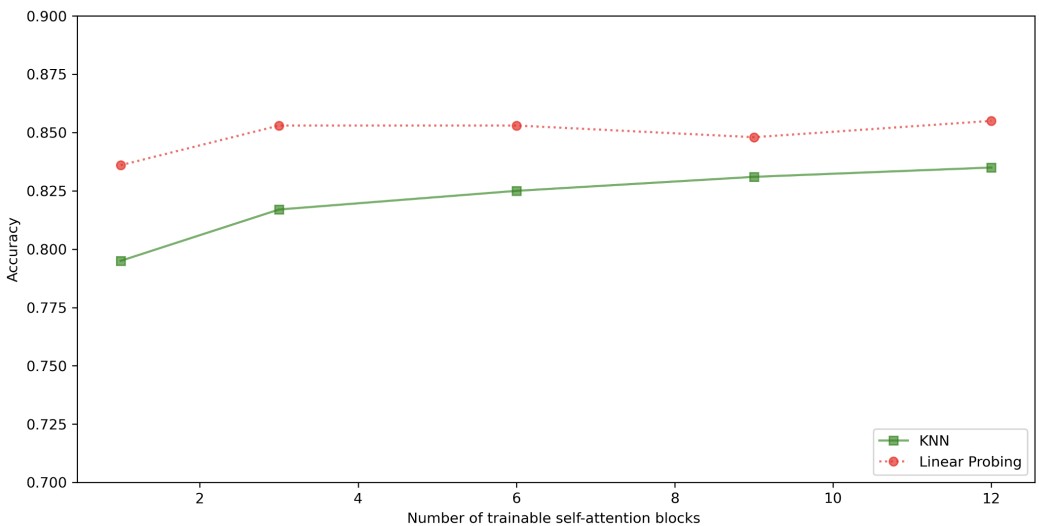

Figure 6: Effect of Model Complexity

## E.3 HARD VS. SOFT ASSIGNMENT

Our multi-stage aggregation uses hard discrete assignments where each token is assigned to exactly one factor group. We ablate this design choice by comparing three assignment mechanisms.

**Three Assignment Strategies** **(1) Standard Softmax (Fully Soft):** Tokens are softly aggregated using standard softmax attention:

$$A = \text{softmax}(\text{scores}), \quad f_{\text{new}} = A \cdot F_{\text{old}} \tag{15}$$

This allows maximum flexibility but permits redundancy (tokens contribute fractionally to multiple groups).

**(2) Gumbel-Softmax (Semi-Soft):** Tokens use Gumbel-Softmax with temperature $\tau = 0.1$ for sharper but still soft assignments:

$$A = \text{gumbel\_softmax}(\text{scores}, \tau), \quad f_{\text{new}} = A \cdot F_{\text{old}} \tag{16}$$

This produces near-discrete assignments while maintaining differentiability.

**(3) Hard (Ours):** We combine Gumbel-Softmax with one-hot encoding using straight-through estimator:

$$\begin{aligned}
A_{\text{soft}} &= \text{gumbel\_softmax}(\text{scores}, \tau) \\
A_{\text{hard}} &= \text{one\_hot}(\arg\max(A_{\text{soft}})) \\
f_{\text{new}} &= (A_{\text{hard}} + A_{\text{soft}} - \text{sg}(A_{\text{soft}})) \cdot F_{\text{old}}
\end{aligned} \tag{17}$$

This ensures each token is assigned to exactly one group (discrete assignment) while maintaining gradient flow during training via the straight-through estimator.

**Experimental Results** Table 17 compares the three strategies on ImageNet-1K with all other settings identical.

Table 17: Comparison of assignment mechanisms in multi-stage aggregation. All results averaged over 3 random seeds.

| Assignment | KNN (%) | Linear (%) | SEPIN@1 | $\|F^T F - I\|$ |
|---|---|---|---|---|
| Standard Softmax (Fully Soft) | $79.8 \pm 0.3$ | $83.2 \pm 0.4$ | $1.52 \pm 0.08$ | 0.34 |
| Gumbel-Softmax (Semi-Soft) | $81.3 \pm 0.2$ | $84.1 \pm 0.3$ | $2.14 \pm 0.09$ | 0.18 |
| **Hard (Ours: Gumbel-Softmax + One-Hot)** | **$84.2 \pm 0.3$** | **$86.0 \pm 0.2$** | **$3.95 \pm 0.12$** | **0.08** |
| Improvement (Hard vs. Soft) | **+4.4%** | **+2.8%** | **2.6×** | **4.2×** |

**Key Observations:**

1. **Sharper assignments improve performance progressively.** Standard softmax $\rightarrow$ Gumbel-Softmax yields +1.5% KNN and 1.4× SEPIN@1. Gumbel-Softmax $\rightarrow$ Hard yields +2.9% KNN and 1.85× SEPIN@1. This demonstrates that discreteness matters: the sharper the assignment, the better the disentanglement.

2. **Hard assignment dramatically improves orthogonality.** The progression in $\|F^T F - I\|$ ($0.34 \rightarrow 0.18 \rightarrow 0.08$) shows that softer assignments lead to more correlated factors, directly contradicting the MVC objective of orthogonal factors. Hard assignment maintains near-orthogonality ($\|F^T F - I\| = 0.08$), which is 4.2× better than standard soft assignment.

3. **Hard assignment achieves best representation quality.** Despite being more constrained (discrete assignments), hard assignment achieves the highest KNN (84.2%) and linear probe (86.0%) accuracy, demonstrating that the enforced disentanglement provides structure that benefits downstream tasks rather than hurting them.

## E.4 HYPERPARAMETER SENSITIVITY

We analyze XTRA's sensitivity to key hyperparameters on ImageNet-1K to demonstrate robustness and provide guidance for practitioners. Table 18 shows results across different hyperparameter values.

Table 18: Hyperparameter sensitivity analysis. XTRA is robust across reasonable ranges, with performance varying by only $\pm 1$–$2\%$ around optimal values. Default settings: $\lambda_{\text{distill}} = 1.0$, $\lambda_{\text{factor}} = 0.45$, $\lambda_{\text{volume}} = 0.05$, $M = 8$ factors, 2 aggregation stages.

| Hyperparameter | Value | KNN (%) | SEPIN@1 |
|---|---|---|---|
| Volume penalty $\lambda_{\text{volume}}$ | 0.01 | 82.1 | 2.87 |
| | 0.03 | 83.8 | 3.64 |
| | **0.05 (default)** | **84.2** | **3.95** |
| | 0.07 | 83.6 | 3.71 |
| | 0.10 | 83.5 | 3.42 |
| Factor loss weight $\lambda_{\text{factor}}$ | 0.25 | 82.4 | 3.28 |
| | 0.35 | 83.7 | 3.82 |
| | **0.45 (default)** | **84.2** | **3.95** |
| | 0.55 | 83.9 | 3.87 |
| | 0.65 | 83.5 | 3.78 |
| Number of factors $M$ | 4 | 81.8 | 2.95 |
| | 6 | 82.8 | 3.21 |
| | **8 (default)** | **84.2** | **3.95** |
| | 10 | 83.1 | 3.67 |
| | 12 | 82.5 | 3.42 |
| Aggregation stages | 0 | 13.9 | 0.51 |
| | 1 | 77.5 | 2.14 |
| | **2 (default)** | **84.2** | **3.95** |
| | 3 | 83.4 | 3.68 |

**Key Findings**   **(1) Robust within reasonable ranges.** Performance varies by only $\pm 1$–$2\%$ across neighboring hyperparameter values, indicating XTRA is not overly sensitive. For example, $\lambda_{\text{volume}} \in [0.03, 0.07]$ all achieve $>83.5\%$ KNN and $>3.6$ SEPIN@1.

**(2) Volume penalty should be small but non-zero.** Too large ($\lambda_{\text{volume}} = 0.10$) over-constrains factors, reducing flexibility (83.5% KNN vs. 84.2% at optimal). Too small ($\lambda_{\text{volume}} = 0.01$) provides insufficient orthogonality enforcement (SEPIN@1 = 2.87 vs. 3.95 at optimal). The optimal range is $[0.03, 0.07]$.

**(3) Factor count $M = 8$ balances expressiveness and efficiency.** Fewer factors ($M = 4$) lack capacity to capture fine-grained parts (SEPIN@1 = 2.95). More factors ($M = 12$) lead to redundancy and harder optimization (KNN = 82.5%). $M = 8$ provides sufficient capacity for part-level decomposition while maintaining tractable optimization.

**(4) Aggregation is critical.** Without aggregation (0 stages), the method completely fails (KNN = 13.9%) due to token collapse (Section 3.2). One stage (77.5%) partially addresses collapse but is insufficient. Two stages (84.2%) provide optimal balance. Three stages (83.4%) over-aggregate, losing fine-grained information.

**(5) Relative weighting matters.** The hierarchy $\lambda_{\text{distill}} > \lambda_{\text{factor}} > \lambda_{\text{volume}}$ (i.e., $1.0 > 0.45 > 0.05$) ensures distillation remains primary, factor learning is auxiliary, and volume constraint is a mild regularizer. This ranking is consistent with the method's design: learn good representations first, then structure them.

# F   FACTOR TOKEN VISUALIZATION

## F.1   MORE RESULTS

In this paper, we consider the initial extra tokens as "mixtures" of semantic contents in the scene. By incorporating the minimum volume constraint and the consistency constraint between the extra tokens and patch tokens, we can generate remarkable attention maps with much finer details while preserving semantic consistency. The main paper showed that the different factor tokens can present

different parts of the object, including high norms Darcet et al. (2024). In this part, we present more results. See Fig. 8 and Fig. 9. In Fig. 8, we show the results on different animals, and the results show the capability of the factor tokens to capture the semantic part. We also show the factor token with a high norm. In Fig. 9, we show the results on other objects, such as no animals, which is easier to capture the whole object. We think it may be because these objects are difficult to disentangle into part-wise properties. This is a potential direction for our future exploration.

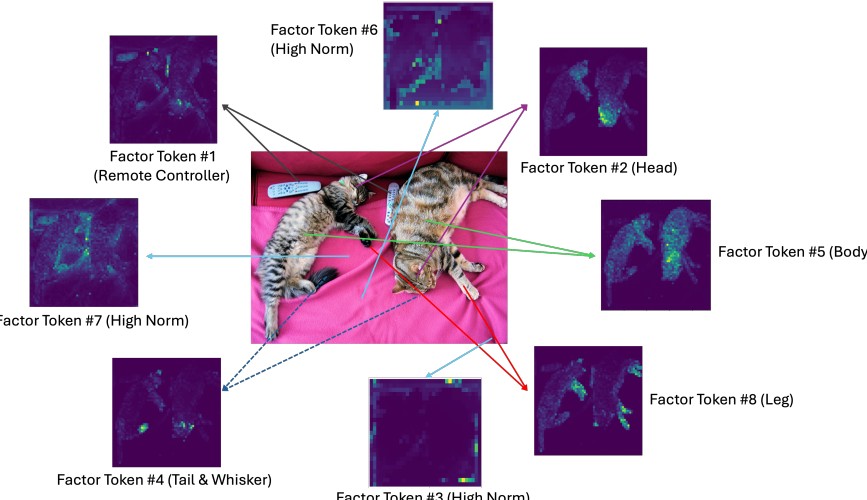

Figure 7: XTRA enables disentangled attention maps pertaining to consistent parts across multiple objects in the scene.

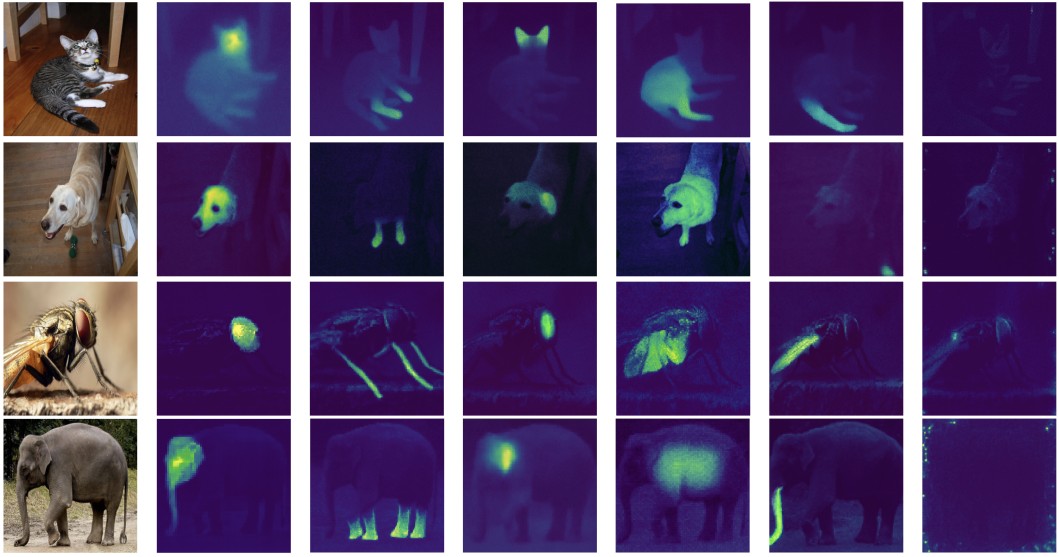

Figure 8: Visualization of Factor Tokens

## F.2 FAILURE CASE ANALYSIS

In Fig.9, airplanes and ambulances exhibit less clear part decomposition, which constitutes a failure case for disentanglement performance. We think some potential reasons cause the failure case: **(1) Rigid objects with uniform appearance:** Unlike animals with distinct part textures (fur patterns, facial features), vehicles have more uniform surfaces. The semantic "parts"

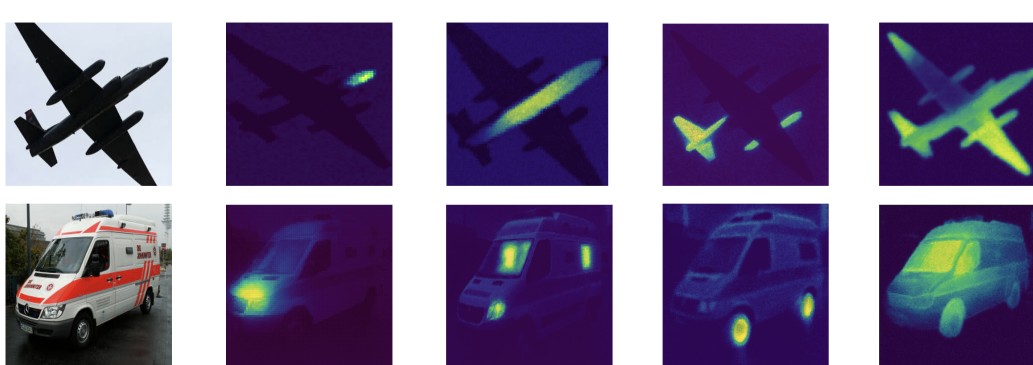

Figure 9: Visualization of Factor Tokens (Other Objects)

(wings, fuselage for planes; body, wheels for ambulances) are less distinct in the learned feature space.**(2) Training data bias:** ImageNet-1K contains more animals than vehicles, potentially biasing part discovery toward biological structures that appear more frequently during training. **(3) Semantic ambiguity:** For vehicles, "parts" (hood, door, wheel) may be less distinct in feature space than animal parts (head, leg, tail) because vehicles have more uniform color/texture (e.g., all parts are painted the same color), less deformable structure (rigid vs. articulated), and less consistent part arrangement (cars vary more in design than animal body plans).

